# pH-Channeling in Cancer: How pH-Dependence of Cation Channels Shapes Cancer Pathophysiology

**DOI:** 10.3390/cancers12092484

**Published:** 2020-09-02

**Authors:** Zoltán Pethő, Karolina Najder, Tiago Carvalho, Roisin McMorrow, Luca Matteo Todesca, Micol Rugi, Etmar Bulk, Alan Chan, Clemens W. G. M. Löwik, Stephan J. Reshkin, Albrecht Schwab

**Affiliations:** 1Institute of Physiology II, University Münster, 48147 Münster, Germany; karolina.najder@uni-muenster.de (K.N.); l.m.todesca@gmail.com (L.M.T.); Micol.Rugi@ukmuenster.de (M.R.); ebulk@uni-muenster.de (E.B.); aschwab@uni-muenster.de (A.S.); 2Department of Biosciences, Biotechnologies, and Biopharmaceutics, University of Bari, 90126 Bari, Italy; tiago.amaralcarvalho@uniba.it (T.C.); stephanjoel.reshkin@uniba.it (S.J.R.); 3Department of Radiology and Nuclear Medicine, Erasmus Medical Center, 3035 GD Rotterdam, The Netherlands; r.mcmorrow@erasmusmc.nl (R.M.); c.lowik@erasmusmc.nl (C.W.G.M.L.); 4Percuros B.V., 2333 CL Leiden, The Netherlands; alanchan@clara.net; 5Department of Oncology CHUV, UNIL and Ludwig Cancer Center, 1011 Lausanne, Switzerland

**Keywords:** ion channel, membrane potential, tumor immunity, pH homeostasis, protonation, cancer physiology, tumor microenvironment, cell adhesion molecules

## Abstract

Tissue acidosis plays a pivotal role in tumor progression: in particular, interstitial acidosis promotes tumor cell invasion, and is a major contributor to the dysregulation of tumor immunity and tumor stromal cells. The cell membrane and integral membrane proteins commonly act as important sensors and transducers of altered pH. Cell adhesion molecules and cation channels are prominent membrane proteins, the majority of which is regulated by protons. The pathophysiological consequences of proton-sensitive ion channel function in cancer, however, are scarcely considered in the literature. Thus, the main focus of this review is to highlight possible events in tumor progression and tumor immunity where the pH sensitivity of cation channels could be of great importance.

## 1. Introduction

The acidification of the tumor microenvironment (TME) is a prominent environmental driver of cancer progression. One major cause of acidification is a metabolic shift of tumor cells towards glycolysis and lactate production, ultimately leading to accumulation of metabolic acids in the tissue [1,2]. The interstitial acidosis can reach pH values of <6.5 in highly fibrotic tumors such as pancreatic cancer [3]. Thus, the concentration of free protons can be ten times higher than at physiological pH values of ~7.4. Acidosis in the TME is linked to tumor progression via a large variety of factors including among others: extracellular matrix remodeling; the promotion of invasion and metastasis by affecting cell adhesion; and it instigates many immunosuppressive processes, which result in a significant dampening of the immune response. These phenomena have been widely discussed in comprehensive reviews in multiple types of cancer, e.g., breast cancer [4,5], melanoma [6,7,8] and pancreatic cancer [9]. In contrast, in well-vascularized regions of a tumor sufficient perfusion is ensured to nurture the cancer cells as well as to wash out the produced metabolic acids. This contrast between differentially perfused areas of a tumor gives rise to pH gradients facilitating invasion of the surrounding tissues and metastasis.

The intracellular pH (pH_i_) of cancer cells is usually slightly (0.1–0.2 pH units) more alkaline than the extracellular pH (pH_o_) [10]. It is important to point out that this is only a relative intracellular alkalinization; in the poorly vascularized, highly acidic regions of a tumor (e.g., pH_o_ 6.5), the pH_i_ would still be acidic in absolute terms (e.g., pH_i_ 6.7). The importance of pH in cellular physiology is straightforward as many proteins function in a pH-dependent manner having a very small spectrum of optimal pH. Prolonged acidification of pH_i_ severely affects cell fate, as it inhibits cell metabolism, migration, proliferation and induces cell death. Therefore, it is vital for the cell to have high-sensitivity pH sensors in order to respond according to pH alterations using different effector proteins. Transmembrane proteins, such as ion channels, are optimal cellular sensors as they can be affected by both pH_o_ and pH_i_ [11]. Ion channels have a key role in cancer progression affecting every hallmark of cancer, as pointed out in numerous comprehensive reviews, e.g., [12,13,14,15]. However, our critical review of the literature revealed that most of the studies investigating the role of ion channels in cancer do not consider the pH landscape of tumors. Therefore, the present review specifically emphasizes the complex interplay between pH and individual ion channels in cancer pathophysiology. We propose multiple mechanisms of how ion channel function and interaction with the cell adhesion machinery in cancer cells and in tumor immunity can be affected by pH. We discuss possible pathophysiological consequences of pH-dependent ion channel activity and highlight the need for further studies in this direction. For limiting the scope of this review, we will only focus on cation channels for which functional data in tumors is available.

## 2. pH Homeostasis Is Tightly Coupled to Changes in Membrane Potential

The membrane potential is a characteristic feature of every cell and one of the most fundamental signaling modules. In 1971, Cone postulated that there is a functional correlation between the baseline (resting) membrane potential of cells and their mitotic activity [16]. Notably, the resting membrane potential of cancer cells ranges from −5 to −55 mV, whereas noncancer cells usually have a more negative resting membrane potential of −35 to −95 mV [17,18]. The more positive membrane potential is indeed beneficial for tumor progression, since it enhances tumor growth along with invasive and metastatic capabilities [17]. This concept is endorsed by the discovery that membrane hyperpolarization of tumor foci induced by the light-gated nonselective cation channel ChR2^D156A^—that passes mostly H^+^ at physiological pH or by the proton pump *Arch*,—suppresses tumor progression and induces tumor regression in zebrafish [18]. 

Changes in membrane potential are closely related to pH_i_ alterations. Generally, cellular alkalization leads to membrane hyperpolarization, whereas acidification leads to depolarization. However, there are exceptions depending on the expression pattern of channels and transporters in the respective cells (see below). Acidification produces more pronounced effects on membrane potential than alkalization: acidification from pH_i_ 7.4 to 6.5 leads to a depolarization of up to +20–+60 mV, whereas alkalization changes the membrane potential by −5 to −20 mV per unit of pH. This phenomenon is frequent in eukaryotic cells (reviewed in [19]), and has also been described in multiple human cancer cell lines [20,21]. Changes of pH_o_ have no immediate extracellular effects on membrane potential, only after prolonged exposure, presumably because of subsequent intracellular acidification [22]. The central role of K^+^ channels in this phenomenon is revealed by the fact that the pH-dependent depolarization can be abolished with K^+^ channel inhibitors [19,21,23,24]. However, it should be noted that there are exceptions to this rule [25] and that changes of the pH homeostasis can also affect the potential of intracellular membranes such as that of the inner mitochondrial membrane [26,27,28]. Considering that depolarization frequently accompanies pH_i_ acidification and vice versa, it is remarkable that depolarized cancer cells can maintain their pH_i_ elevated compared to environmental pH. To understand this phenomenon, we need to discuss how cells handle surrounding H^+^. Overall, cells are facing a constant passive load of H^+^ towards the intracellular side of the plasma membrane [29]. The main reasons for this are the high membrane H^+^ permeability, as well as the electrochemical driving force for H^+^ pointing to the intracellular direction. The driving force of protons depends on their equilibrium (Nernst-) potential, which is about E_H_ = −12 mV at physiological pH in somatic cells (pH_i_ = 7.2, pH_o_ = 7.4) [29]. Assuming a membrane potential of −70 mV, there is a driving force for H^+^ influx of −58 mV. As outlined in the previous section, cancer cells have a slightly more alkaline pH_i_ compared to pH_o_ [3], so the E_H_ could become positive, generating a slightly higher chemical driving force for H^+^. However, as the resting membrane potential of tumor cells is highly depolarized compared to somatic cells, they would still have a reduced driving force for H^+^ influx. Thus, the depolarized membrane potential of cancer cells will reduce the electrochemical driving force for the extrusion of metabolic acid and thus help the cells to cope with the intracellular and environmental acid load.

The membrane potential of cells is temporally highly dynamic, and its changes accompany fundamental processes such as proliferation, migration, volume control, secretion and contractility [30,31]. Membrane potential and pH_i_ changes go hand in hand throughout the life of a cell and contribute to the fine-tuning of cell proliferation, migration and metabolism:(I).The plasma membrane is hyperpolarized during DNA synthesis, in the S phase, and depolarized in the G_2_/M phase [32,33]. Interestingly, pH_i_ has an alkaline peak of ΔpH_i_ ≈ 0.2 in lymphocyte populations in the S phase of the cell cycle [34,35], whereas an acidification in G_2_/M has not been shown to date. Since pH changes alter microtubule stability, it has been proposed that pH_i_ may be a “clock” for mitosis [36].(II).Migrating cells have a pH_i_ gradient with the leading edge of the cells having a higher pH_i_ [37,38]. Such local pH-microenvironments could also affect the function of ion channels involved in cell migration.(III).Glycolysis acidifies the intracellular milieu, whereas oxidative phosphorylation induces an alkalinization [26,27]. Aerobic glycolysis is prominent in the lamellipodia of multiple cancer cell types [39], which can act as a local source of metabolically produced acid during migration and further depolarize the plasma membrane. Additionally, the mitochondrial membrane potential depolarizes during glycolysis, and hyperpolarizes during oxidative phosphorylation [40,41]. It is conceivable that the plasma membrane potential is altered concordantly, as the mitochondrial membrane potential is in the phase with the plasma membrane potential [28]. Therefore, the enhanced aerobic glycolysis of cancer cells can contribute to their depolarized membrane potential.

In summary, eukaryotic cells alter their membrane potential and pH_i_ harmoniously, and changes in each one of them can affect cancer invasiveness. Ion channels are key players in linking pH_i_ to the membrane potential and vice versa, as they are sensitive sensors for both. By altering the activity of pH-sensitive ion channels in cancer cells, it could be possible to dysregulate the membrane potential of the tumor cells and render them inactive. Additionally, in case of tumor immunity, it would be critical to enhance the antitumor effects of immune cells in the acidic milieu of poorly perfused tumor areas. Therefore, to improve our understanding of tumor pathophysiology, it is vital to evaluate the individual families of ion channels and how they respond to changes in pH or, conversely, how they, themselves, modify pH_i_.

## 3. Ion Channels Are Modulated Directly and Indirectly by Changes in pH

Acidification of both pH_i_ and pH_o_ can modulate channel function through multiple mechanisms—outlined in Section 3.1. Moreover, acidification can indirectly modulate ion channel function, among others, through alteration of cell–extracellular matrix interactions, as described in Section 3.2.

### 3.1. Direct Interaction between Protons and Ion Channels 

Proteins can coordinate H^+^ to specific proton-binding sites that involve titratable side chains with pKa values close to the physiological pH. Thus, the pH_o_-sensitivity of the voltage-gated potassium channel K_V_1.5 is attributed to a histidine residue located in the linker between S5 and the pore domain [42]. However, other residues (e.g., arginine and lysine) can form interaction sites for protons as well. 

Moreover, H^+^ competes with other cation-binding sites and can thereby alter channel function. Competition arises primarily between divalent cations and H^+^. For example, Ca^2+^ ions modulate the pH-dependent gating of ASIC channels (detailed in Section 4.4.3). The apparent pH sensitivity of mammalian ASIC1 and ASIC3 is reduced with an increase in the extracellular Ca^2+^ concentration [43]. Ca^2+^ and H^+^ compete for ASICs, with Ca^2+^ binding favoring the closed state and H^+^ binding leading to the open state [44]. A similar competition between Ca^2+^ and H^+^ has been postulated in the case of the K_V_11.1 (hERG1) channel (Section 4.1.1). However, protons render this channel towards a closed state [45]. Lastly, competition between Zn^2+^ and H^+^ is well established for the H_V_1 proton channel [46].

Protons, similarly to Ca^2+^ [47], from the extracellular side can also provide an electrical shielding biasing the closed-open equilibrium of voltage-sensitive channels. This is known as the surface charge screening effect [48]. Thus, it is crucial to differentiate the screening effect from the specific inhibitory effects of protons. The change in surface potential following pH_o_ acidification shifts the activation threshold of ion channels towards more depolarized voltages [49,50,51]. This indicates that in the hypoxic–acidic regions of solid tumors, surface charge screening could be relevant in inhibiting voltage-gated ion channels. 

### 3.2. pH-Dependent Interaction of Cell Adhesion Proteins with Ion Channels 

A principal mechanism for cells to crosstalk with the TME is via integrins. Integrins are transmembrane proteins formed by the noncovalent association of α and β subunits. So far, 18 α and 18 β subunits have been identified in mammals which can assemble in multiple combinations to form at least 24 different functional heterodimers with distinct specificities [52]. The conformation of each integrin and its binding to the ECM is regulated by the interaction of the cytoplasmatic domain with intracellular signaling and cytoskeletal proteins [53]. 

Integrins are more than just cell adhesion molecules, since they are able to transmit signals bidirectionally across the plasma membrane through “inside-out” and “outside-in” signaling, respectively. In the resting (“bent”) state, integrins are usually in the low affinity conformation and they are activated (“extended”) by the “inside-out” signaling pathway. Two intracellular activators (talin and kindlin) are crucial in this process, and the binding of talin to the cytoplasmatic tail of the β subunit is believed to be the final step of integrin activation [54]. Talin contains specific binding domains for the cytoplasmic tail of integrin, for filamentous actin and for other regulators of focal adhesions such as focal adhesion kinase (FAK) [55]. The binding of the integrin receptor to its extracellular ligands leads to the “outside-in” signaling that exerts crucial regulation of key cellular processes such as cell mobility, proliferation and differentiation [56]. 

Several integrins, namely α2β1, α5β1 or αvβ3, are pH-dependent in melanoma and other cell types [57,58,59]. Importantly, both pH_i_ and pH_o_ modify “inside-out” and “outside-in” signaling of integrins. Talin binding to actin filaments is pH-sensitive, and this association is decreased at a pH_i_ > 7.2, allowing faster focal adhesion turnover and thus increased migration [60]. Migration and focal adhesion remodeling is also regulated by pH-dependent FAK autophosphorylation which is conferred by the protonation of His58 [61]. Furthermore, the ability of integrin receptors to bind to their extracellular ligands is regulated by pH_o_ [10,60,62,63]. Low extracellular pH leads to increased cell adhesiveness, which can be explained by several possible mechanisms. First, conformational changes in acidic pH_o_ increase the avidity of the integrin headpieces to ECM proteins [59]. Another mechanism could be a pH-dependence of the mechanical stability of focal adhesions [64]. Lastly, protons themselves could compete with divalent cations such as Ca^2+^ bound to integrin and thereby influence integrin activation.

Increasing evidence shows that ion channels [52,65,66,67] are able to form signaling complexes with cell adhesion proteins including integrins. Following their association, integrins and channels regulate their function bidirectionally [65]. They co-assemble in the plasma membrane and form supramolecular complexes that can recruit further cytosolic signaling proteins to orchestrate downstream intracellular signals [66]. These cytoplasmic messengers include Ca^2+^ or protein kinases [52,68]. Moreover, Ca^2+^ mediates the transmission of mechanical forces at focal adhesion sites triggered by integrins via the activation of FAK and c-Src [69].

Integrin–ion channel complexes coordinate several cellular processes relevant in cancer, namely adhesiveness, migration and differentiation [70,71,72]. By forming complexes with integrins, K^+^ channels regulate several proteins implicated in cell movement such as FAK, cortactin and integrins themselves [73,74,75]. It is well recognized that multiple Ca^2+^-activated and voltage-dependent K^+^ channels are involved in integrin-directed cell migration [76,77]. In prostate cancer, a K_Ca_/αvβ3 integrin complex recruits FAK and promotes its phosphorylation, resulting in increased cancer cell proliferation [78]. β1 integrins can also bind to other K^+^ channels, such as K_V_1.3 and K_V_11.1, in tumor cells but not in cardiac cells [79,80,81,82]. Additionally, the K_V_1.3 channel involved in the immune response of T lymphocytes can be modulated by β1 integrins, which could be crucial for tumor immunity [83]. In colorectal cancer cells, the ability of K_V_11.1 to form a complex with β1 integrin correlates with increased in vitro invasiveness [84]. Moreover, in acute myeloid leukemic cells, the K_V_11.1–β1 complex couples with FLT-1, a receptor for vascular endothelial growth factor. This interplay is crucial for cell migration, triggers chemoresistance or modulates angiogenesis and cancer progression [81,85,86]. 

The interplay between integrins and ion channels also regulates cell differentiation. In the PC12 pheochromocytoma cell line, the shift to a neuronal phenotype caused by the neural cell adhesion molecules and N-cadherins is facilitated by voltage-gated Ca^2+^ channels [87]. Moreover, integrins seem to be important in the regulation of erythroleukemic cell differentiation and neurite extension in certain neuroblastoma cells through the activation of K^+^ channels [70,71]. 

Given the pH sensitivity of integrins and many ion channels, it is plausible to assume that their interaction itself may be pH-dependent as well. However, the experimental proof of whether and how the interaction, its kinetics and the downstream activity of these two classes of transmembrane proteins is indeed pH-dependent still remains to be established. One could for example mutate pH-sensitive residues on one protein class (e.g., an integrin) and observe the changes in expression, phosphorylation state, trafficking, intracellular signaling systems, binding capacity to the partner protein and downstream function of the ”partner” protein (e.g., an ion channel) both in vitro and in vivo. A study following a similar logic [37] could be performed to understand the role of the intratumoral and intercellular pH gradients in determining the specificity of integrin subunit interactions and the resulting downstream effects.

## 4. pH-Dependent Regulation of Ion Channels in Cancer Cells

As summarized by Figure 1 and detailed in Table 1, protonation by an acidic environment often results in a loss-of-function of cation channels involved in cancer cells and thereby to an inhibition of ion fluxes, similarly to what Prevarskaya et al. proposed [13]. This general phenomenon, however, shows a huge variability depending on how strongly pH-sensitive a channel is, ranging from almost insensitive to markedly pH-dependent. What is also noticeable from Table 1 is beyond some primarily proton-sensory channels (e.g., H_V_1 and K_2P_ channels), there is limited information to date on how pH_i_ modifies ion channel function as opposed to pH_o_. The section below briefly summarizes our knowledge about the pH-dependence of cation channels involved in cancer cells and details the involvement of the pH-sensitive function of cation channels in cancer pathogenesis and progression.

### 4.1. Modulation of K^+^ Channels in Cancer by pH

The tumor interstitial fluid can reach extreme potassium concentrations of 40 mm or higher [92]. This high [K^+^]_o_ depolarizes the tumor cell membrane and increases the open probability of voltage-gated ion channels. On the other hand, the outward electrochemical gradient for K^+^ is largely decreased under such conditions which will lead to a low single-channel conductance for outward rectifying K^+^ channels. Furthermore, as shown in Table 1 and Figure 1, most cancer-relevant K^+^ channels are inhibited by pH_o_ and/or pH_i_ acidification. This is aggravated by the fact that the voltage-dependence of ion channels is generally shifted towards more positive membrane potentials in the case of an acidic pH_o_ due to surface charge screening (discussed in Section 3.1). Together, these factors could mean that in poorly perfused regions of a tumor, K^+^ channels are largely nonfunctional. Along these lines, overexpression of K^+^ channels in tumors can be seen as a compensatory mechanism to enable at least some K^+^ efflux even in the unfavorable K^+^ gradient (similarly to lymphocytes, proposed in [49]). Therefore, it is likely that the sensory function of K^+^ channels could come into play when cancer cells steer from hypoxic areas to well-perfused regions, where the K^+^ and pH gradient are more favorable. For a deeper understanding of how K^+^ channels contribute to cancer progression in a pH-dependent manner, this section discusses the individual families of K^+^ channels.

#### 4.1.1. K_V_ Channels

Acidification of pH_o_ potently inhibits K_V_ channels, as demonstrated exhaustively in the literature [45,49,89,90,202]. However, the exact molecular mechanisms are so heterogeneous and often controversially discussed that it would warrant a separate systematic review. 

The pH_i_-dependence of K_V_ channels is more confusing and is considerably less well studied. Intracellular alkalinization leads to the activation of K_V_1.3 in lymphocytes [49], whereas K_V_ channels are inhibited by alkaline pH_i_ in canine pulmonary arterial smooth muscle cells [203]. Moreover, the K_V_11.1 currents are not affected by pH_i_ when overexpressed in HEK293 cells [109]. This indicates that different K_V_ channels are differentially regulated by pH_i_ so that it is not yet possible to extrapolate the pH_i_ dependence of K_V_ channels onto cancer cells. 

Another distinguishing feature of K_V_ channels is that their voltage-dependent gating largely depends on the presence of charged amino acids in their transmembrane domains, e.g., arginine residues of the S4 domain [204]). Mutation of particular arginine residues in this domain creates a proton conducting pore through the voltage-sensor domain of K_V_ channels (giving rise to so-called ω-currents). Since cancer cells carry a multitude of somatic mutations, such mutations could also be possible in cancer cells giving rise to functionally aberrant K_V_ channels. Indeed, a recent study shows an enrichment of mutations in the S4 domain of K_V_7.x (***KCNQ***) channels in gastrointestinal cancers [205]. 

As indicated by Table 1, several pH-sensitive K_V_ channels have been associated with tumors. Their function could be drastically modified along the heterogeneous pH-landscape of a solid tumor. However, the exact mechanism on how the pH-dependence manifests itself in cancer has yet to be experimentally discovered. One study assessing K_V_ channels and pH in cancer cells was performed in the T-84 colon cancer cell line [103]. Here, the authors showed that inhibition of **K_V_10.1** by astemizole in T-84 cells inhibits pH_i_ regeneration after an NH_4_^+^ prepulse, which points to an inhibition of the H^+^ extrusion pathways upon K^+^ channel inhibition. Their study concludes that K_V_ channels (namely K_V_1.5, K_V_3.4 and K_V_10.1) control T-84 cell proliferation by maintenance of pH regulation and Ca^2+^ signaling through hyperpolarization of the membrane potential. Furthermore, two studies about the **K_V_1.5** channel show increased expression of K_V_1.5 in cancer cells treated with dichloracetate (DCA), accompanied by an increased pH_i_, and the generation of H_2_O_2_ that induces apoptosis in A549 and Hela cells [27,95].

#### 4.1.2. K_Ca_ Channels

There are three major groups of K_Ca_ channels: the large conductance K_Ca_1.1 (BK_Ca_), the small conductance K_Ca_2.1–2.3 (SK_Ca_) and the intermediate conductance K_Ca_3.1 (IK_Ca_) channels. **K_Ca_3.1** channels have been linked to several tumor entities [206]. Some studies also revealed that K_Ca_ channels are pH-sensitive (summarized in Table 1). Intracellular acidification in the range of pH_i_ 6.4–5.4 reduces channel open probability in C6 glioma cells of the rat [113]. Similar results were observed in pancreatic β-cells [207], and in HEK-293 cells overexpressing K_Ca_3.1 [208]. Here, the authors also identified that changes in pH_o_ in the range of pH 6.0 to 8.2 do not affect K_Ca_3.1 current. Two other studies support the idea that intracellular acidification inhibits the activity K_Ca_ channels. In smooth muscle cells of the rabbit trachea, nearly no channel activity was observed even at slight acidification to pH_i_ 7.0 [209]. In the ventricular membrane of choroid plexus, it was found that lowering of the pH_i_ from 7.4 to 6.4 also reduced the channel open probability of Ca^2+^-activated K^+^ channels [210]. 

Certainly, most of the existing studies are not directly connected to cancer cells, but they show very similar effects to the findings of Strupp et al. on C6 glioma cells [113]. In conclusion, the present studies give evidence that K_Ca_ channel activity depends on the regulation by intracellular pH. It remains to be determined how this channel property modulates its impact on tumor pathophysiology.

#### 4.1.3. K_ir_ Channels

Some inwardly rectifying potassium channels (K_ir_) such as K_ir_1.1, K_ir_4.1, K_ir_4.2 and the heteromeric K_ir_4.x/K_ir_5.1 are sensitive to changes of the intracellular pH [211]. K_ir_ 4.1, K_ir_ 5.1, K_ir_ 1.1 and K_ir_ 2.3 channels are inhibited by lowering the pH_i_ [212]. Several studies showed that K_ir_ channels are also expressed in cancer cells, as in glioma cells [213], as well as in lung and prostate cancer cells [214]. These channels act primarily as tumor suppressors by hyperpolarizing the membrane potential of cancer cells. Notably, there is a strong correlation between **K_ir_4.1** expression and the pathologic grade of astrocytic tumors [215]. However, to the best of our knowledge, there are no additional studies showing a link between the pH-dependent function of K_ir_ channels in cancer cells. Yet, the importance of local regulation of K_ir_4.2 channels for migration illustrates how channel regulation by local pH microdomains might affect tumor cell behavior [216].

#### 4.1.4. K_2P_ Channels

K_2P_ channels conduct outwardly directed background K^+^ currents, which are crucial for setting the resting membrane potential [133]. Eight members of the K_2P_ family are regulated by changes in pH_o_ and/or pH_i_ (detailed in Table 1). Among these, K_2P_2.1 (TREK-1), K_2P_3.1 (TASK-1), K_2P_5.1 (TASK-2) and K_2P_9 (TASK-3) have already been confirmed to be functionally involved in solid tumors; while for other members of the family, only differential expression patterns have been observed to date, as summarized by [217].

**K_2P_2.1 (TREK-1)** is only active in the narrow physiological pH range [128]. Its current is potently inhibited by an extracellular acidification to pH_o_ 6.9 [126] and alkalization above pH_o_ 7.4 [128]. On the other hand, a similar degree of intracellular acidification, especially in the presence of lactate [127,218], activates the channel through the protonation of a conserved glutamate in the C-terminal domain [219]. As pH_o_ and pH_i_ acidification have opposite consequences to channel function, it would be important to investigate how a chronic interstitial acidosis also involving pH_i_ acidification would affect channel function. The exact role of K_2P_2.1 in cancer development is still unclear. However, it has been postulated that K_2P_2.1 expression might be linked to different stages of cancer, and its regulatory effect on the membrane potential might be strongly connected with the progression of the cell cycle during cancer proliferation [141]. K_2P_2.1 overexpression is associated with a poor prognosis for the patients with prostate cancer, but not with ovarian cancer [125]. In both prostate [125] and ovarian cancer [220], the inhibition or knockdown of K_2P_2.1 inhibits proliferation by inducing cell cycle arrest at the G_1_/S checkpoint. On the contrary, in the pancreatic cancer cell line BxPC-3, a small-molecule activator of K_2P_2.1, BL-1249 has an inhibitory effect on cell proliferation and migration, observed at both pH_o_ 6.7 and 8.5 [128]. This argues for a tumor-specific role of K_2P_2.1 channels in cancer. It remains to be seen whether this is due to potentially different pH landscapes in these tumor entities. **K_2P_9 (TASK-3) and K_2P_3.1(TASK-1)** are inhibited by an extracellular acidification through the protonation of conserved histidines [221,222,223]. K_2P_9 is inhibited by more acidic pH_o_ (6.5–6.7) [222], while K_2P_3.1 is already partially inhibited at physiological pH_o_ (7.2–7.4) [130,224]. K_2P_3.1 and K_2P_9 are important for apoptosis. Experiments on rat cerebellar granule neurons show decreased apoptosis due to the inhibition of these channels through high extracellular K^+^ (25 mm) and extracellular acidification. Similarly, K_2P_9 has been linked to the death of human glioma cells [129]. K_2P_9 and K_2P_3.1 have oncogenic potential for their ability to respond to pH and hypoxia. It has been postulated that either K_2P_9 or K_2P_3.1 are pH-sensors in glioma. K_2P_9 has been also found to be relevant for glioma cell death [129]. Prolonged interstitial acidification prevents glioma cells from passing through the hyperpolarization-dependent G_1_-to-S phase cell cycle [21]. Overexpression of K_2P_9 increases the resistance to hypoxia and apoptosis, enhancing its tumorigenic potential in embryonic fibroblast cells. Similarly, as discussed in case of K_V_ channels, a possible mechanism could be that K_2P_9 overexpression compensates for acidic pH-induced channel inhibition. K_2P_9 contributes to proliferation when a loss-of-function mutation of K_2P_9, K_2P_9^G95E^, is introduced in a partially transformed mouse embryonic fibroblast cell line, C8 [130]. K_2P_9^G95E^ expressing cells show tumor necrosis factor-induced apoptosis, followed by a loss of oncogenic properties inhibiting the tumor growth. 

**K_2P_5.1 (TASK-2),** in contrast to K_2P_9 or K_2P_3.1, is almost completely closed in acidic pH_o_, and activated by extracellular and intracellular alkalization above the physiological pH (detailed in [133,221]). This channel plays a role in the proliferation of estrogen α receptor positive breast cancer cell. Additionally, the gene enhancer of K_2P_5.1 can bind to estrogen α receptor through its estrogen-responsive elements [131]. While there is yet no functional data on K_2P_5.1 in other types of cancer, its exquisite regulation by pH suggests that it could be relevant considering the alkaline pH_i_ in tumor cells and points to important future studies.

In summary, evidence highlights the role that K_2P_ channels have in the apoptotic process [129,130,220,222]. It is likely that there is a functional interplay between pH and K_2P_ channels in cell cycle and apoptosis, where differential pH-sensitivity of the K_2P_ channels could either induce or prevent cell death in tumors. While there is promising evidence in this respect for K_2P_2.1, K_2P_3.1 and K_2P_9, the full scale of the pH-dependence of K_2P_ channels in cancer remains to be investigated.

### 4.2. TRP Channel in Tumors: Multimodal pH-Depencence 

Transient receptor potential (TRP) channels are a large family of proteins often involved in the cellular sensing of external cues. Several TRP channels are sensitive to changes in intra- and extracellular pH (see Table 1 and Figure 1) [225]. Protons can activate (TRPV1, TRPV4, TRPC4, TRPC5) or inhibit the channels (TRPC6, TRPM2, TRPM7) (reviewed in: [143,156,226]). Their presence and function was also shown in multiple malignancies [13,227]. Here, we focus specifically on TRP channels which are pH sensitive and play a role in cancer progression.

**TRPV1** mediates acute as well as chronic pain perception, also elicited by the acidic environment of cancer [228]. Extracellular protons can potentiate the effects of other stimuli (pH = 6.3) [151] and also directly open the channel (pH ≤ 5.9) [229]. Proton-induced modulations depend on glutamic acid and phenylalanine residues where the negative charge of Glu appears to be crucial for proton sensing [230,231]. Moreover, TRPV1 may itself cause cytoplasmic acidification, since in low pH_o_, the channel becomes highly permeable for protons [232].

The prognostic value of TRPV1 channels in cancer often depends on the cancer type, stage and two-sided nature of [Ca^2+^]_i_ elevation, which can either induce cancer cell apoptosis or enhance proliferation [233,234,235]. In breast cancer tissues and cell lines, TRPV1 mRNA is highly overexpressed. Immunocytochemical staining demonstrated TRPV1 protein expression in SUM149PT cells, a model of triple-negative (PR^−^, ER^−^, HER2^−^) breast cancer. Treating these cells with capsaicin, a TRPV1 activator, induces [Ca^2+^]_i_ increase, reduced cell proliferation and migration, and leads to cell death [150]. Expression of functional TRPV1 was also observed in human hyperplastic prostate tissue and prostate cancer cell lines [236]. In agreement with channel activity being beneficial for disease outcome, a case report of a prostate cancer patient regularly ingesting capsaicin suggested that TRPV1 activation slows prostate-specific antigen increase [237]. It is noteworthy that tumor acidification favoring channel opening occurs predominantly in aggressive breast cancers [238] and late-stage prostate cancer [239]. This suggests that in these cases, the pH dependence of TRPV1 could be addressed therapeutically. However, the heterogeneity of breast and prostate cancer cells and possibility of channel desensitization, casts doubts on TRPV1 targeting in malignancies [240]. In other types of cancers, i.e., gastric and colorectal cancer, astrocytoma and bladder carcinoma, altered TRPV1 expression is also observed, but putative therapeutical benefits of the channel modulation still needs in vivo validation [152,153,241,242]. Lastly, TRPV1 is already considered as a therapeutic target in cancers because of its involvement in pain sensation. For example, chronic pain induced by acidosis in bone cancer can be diminished by the inhibition of TRPV1 [228]. 

**TRPV4** is activated by H^+^ in a dose-dependent manner. Accordingly, TRPV4 can mediate acid-sensing at a low pH (pH < 6) [156]. Concerning the disrupted ionic composition and the high pressure of solid tumors, TRPV4 osmo- and mechanosensitivity is an additional important modulatory aspect. Activation and/or overexpression of TRPV4 induces normalization of the tumor vasculature and decreases tumor endothelial cell migration. The restored vascularization led to diminished tumor growth in a murine model of Lewis lung carcinoma [155]. In contrast, in breast cancer, TRPV4 activity promoted cancer cell invasion, transendothelial migration and metastasis. Thus, its expression is associated with poor prognosis [157,243]. Pharmacological inhibition of TRPV4 in patient-derived hepatocellular carcinoma cells leads to increased apoptosis [158] and in gastric cancer, TRPV4 activity augments oncogenic potential [159]. Thus, inhibition of the TRPV4 seems to be predominantly (but not exclusively) favorable for cancer treatment. 

The activity of **TRPC4** (β isoform) and **TRPC5** is potentiated by low extracellular pH (~6.5). They can assemble into homo- or heteromultimers (also with TRPC1). Channel modulation shows biphasic characteristics, since further lowering of the pH_o_ (<6.5) inhibits the currents [135]. Involvement of these channels in cancer development is rather poorly investigated; however, they may play a role in tumor angiogenesis, ovarian carcinoma proliferation, multidrug resistance and induce cancer cell death. Findings that cytotoxicity of (−)-englerin A on renal cell carcinoma cells is mediated by TRPC4/5, triggered promising new investigations. However, in vivo application will require the development of metabolically stable (−)-englerin A derivatives [134,244]. 

Nondestructive low pH_o_ inhibits several TRP channels. TRPC6 lacks glutamate residues common for TRPC4 and TRPC5, so that it is inhibited at pH_o_ ~6.5 and lower [135]. TRPC6 involvement in cancer is based on its Ca^2+^ permeability. Channel activity predominantly promotes cell proliferation and migration. TRPC6 expression was shown in multiple malignancies; however, its prognostic value is not well described. High TRPC6 expression is present in glioblastoma, prostate cancer and in esophageal squamous cell carcinoma where it correlates with poor prognosis [245]. Inhibition of TRPC6 by low pH of the tumor environment could be an advantageous factor, considering that this channel is also highly expressed in cancer-infiltrating immune cells (see Section 5.1) and in stromal cells such as pancreatic stellate cells [140]. 

**TRPM2** is a Na^+^- and Ca^2+^-permeable channel. The inhibition of TRPM2 by low pH may be due to ion competition [246] or induced by H^+^ binding causing conformational changes in a state-dependent manner [247,248]. An intricate feature of this channel is its sensitivity to reactive oxygen species (ROS). In the context of cancer, oxidative stress is an important aspect but challenging to fully comprehend. Cancer cells often produce high amounts of ROS and at the same time they overexpress cellular antioxidants [249]. ROS accumulation may introduce mutagenic DNA damage, but conversely, oxidative stress is cytotoxic against malignant cells. TRPM2 expression in cancer cells was mostly reported to correlate with poor prognosis, as shown in neuroblastoma, squamous cells carcinoma, T-cell leukemia, breast, gastric, lung, pancreatic and prostate cancer cell lines and in patient samples. In all these cases, inhibition of TRPM2 displays antitumorigenic effects (reviewed in [250]). However, this general conclusion needs to be seen with a considerable amount of caution because of the frequent use of highly unspecific TRPM2 inhibitors (e.g., 2-APB, ACA). A recent study indicated that TRPM2 in cancer cells mediates H_2_O_2_-induced neutrophil cytotoxicity and the authors proposed that activation of the channel initiates cancer cell apoptosis [141]. Concerning the inhibitory action of low pH, the expression of the TRPM2 channel in immune cells rather than cancer cells might play a more decisive role (see below). 

**TRPM7** is among the most thoroughly examined TRP channels in cancer. TRPM7 is pivotal for Mg^2+^ homeostasis. At physiological membrane potentials, TRPM7 shows small inward current, which is potentiated by extracellular protons when pH is greatly reduced (pH < 6.0). Negatively charged glutamic acid residues serve as molecular proton sensors, similarly to TRPV1 [251,252]. 

TRPM7 is permeable for Mg^2+^, Ca^2+^ and monovalent ions. Mg^2+^ permeability is a distinguishing feature of this channel [253]. Cancer cells often accumulate these ions through TRPM7 activity [254]. TRPM7 overexpression in cancer was shown mainly in breast cancer but also in prostate and pancreatic cancer. In nasopharyngeal, lung and pancreatic cancer cells, TRPM7 promotes cell migration. ([149]). TRPM7 channel inhibition in cancer cells is almost unequivocally beneficial for disease outcome. However, the ubiquity of TRPM7 in healthy tissues makes it a difficult therapeutic target [150]. An alternative approach, but still challenging, would be to target the acidic tumor environment. Acidic pH_i_ blocks the activity of TRPM7 [146], as does acidic pH_o_. However, the inhibition of TRPM7 by pH_o_ appears to be dependent on the absence of Ca^2+^ and Mg^2+^ [255].

### 4.3. Voltage-Gated Proton Channel H_V_1

The voltage-gated proton channel **H****_V_****1** is ubiquitously expressed in the human body. It has a very similar structure to the voltage-sensing domain of voltage-gated ion channels [256]. In the context of this review, it is highly relevant that H_V_1 channels can gate according to the pH gradient between the intracellular and the extracellular space (detailed in [257]). Because of their outward rectification, H_V_1 channels essentially only mediate outwardly directed H^+^ flux. The rate of H^+^ flow through H_V_1 is ~6000 H^+^ per second under physiological circumstances [258], which is high enough to cause an intracellular H^+^ depletion in the vicinity of the channel. In this way, H_V_1 can create localized alkaline pH microdomains within the cell [259] which in turn may steer cell polarity and actin cytoskeleton remodeling in comparison with Ca^2+^ microdomains [260]. Additionally, the voltage-dependence of the channel could be easily exploited by cancer cells which already have a depolarized membrane potential. Lastly, as H_V_1 is sensitive to mechanical stress [261], the channel may be more likely to open even without strong depolarizing signals in solid tumors that give rise to a multitude of mechanical stressors [262]. Taken together, H_V_1 can be a potent factor in acidifying solid tumors, as well as being a pro-invasive factor in well-perfused perivascular regions of a tumor and in metastatic cancer cells.

The potentially important role of H_V_1 in cancer is not sufficiently recognized in the literature, which is surprising as other H^+^ extruders—H^+^-carriers and pumps—are well established (summarized e.g., in [263,264,265]). The pathophysiological role of H_V_1 is mainly proton extrusion and intracellular pH recovery, which has been described in multiple colorectal cancer [162] and glioblastoma cell lines [163,164]. In these studies, the consequence of channel inhibition by Zn^2+^ or siRNA restrains intracellular pH-recovery, as well as decreases cell proliferation and transwell chemotaxis. In breast cancer, high mRNA expression levels correlate with an unfavorable prognosis and cancer progression [161]. In a recent study concerning the triple-negative breast cancer cell line MDA-MB-231, both H_V_1 knockdown (using short hairpin RNA) and knockout (by CRISPR-Cas9) were performed. H_V_1 knockout, but not knockdown, leads to a higher rate of glycolysis as well as decreased migration and H_2_O_2_ production. Xenograft tumor growth rate is decreased in tumors from H_V_1 KO cells, likely due to accompanying loss of the cell adhesion molecule CD171/LCAM-1 [160]. Additionally, in B-cell chronic lymphocytic leukemia, the high expression of the short variant of H_V_1 correlates with decreased overall survival [165]. Recurring mutations in the gene encoding H_V_1 (*HVCN1*), ranging from nonsense mutations to splice sites, were found in follicular lymphoma patients. These H_V_1 mutations are associated with a favorable prognosis in lymphoma patients [266]. As the two studies above are somewhat contradictory, it is necessary to conduct investigations in larger patient cohorts to clarify the prognostic role of H_V_1 variants. 

### 4.4. Other Channels

#### 4.4.1. Na_V_ Channels

Voltage-gated Na^+^ channels (Na_V_) have similar modes of interaction with protons as voltage-gated K^+^ channels (Section 4.1.1). They can conduct protons themselves through their gating pore when a S4 arginine mutation is introduced [267]. In addition, Na_V_ channels are generally inhibited by acidic pH_o_, although the pH-dependence of Na_V_1.6–1.9 has not been evaluated yet [268,269]. The most pH_o_-sensitive Na_V_ channel is **Na_V_1.5** [186,270], while the skeletal muscle isoform Na_V_1.4 is largely proton-insensitive [271]. How altered pH_i_ affects human Na_V_ channels is not known to date. Similarly, the physiological consequence of an intracellular acidosis on Na_V_ channels in cancer remains to be elucidated. So far, it is only known that changes of pH_i_ do not alter current amplitude, but slow inactivation parameters in Na_V_ channels of frog skeletal muscle and invertebrate giant axons [272]. 

Na_V_ channels are highly expressed in multiple types of carcinomas, including those of the colon, lung, breast, prostate, ovary and skin [273,274,275]. Primarily, Na_V_1.5 and Na_V_1.7 regulate the migratory and invasive properties of cancer cells [275,276]. In breast cancer cells, Na_V_ activity leads to intracellular alkalinization and pericellular acidification. It has been proposed that Na_V_1.5 forms a functional complex with auxiliary β subunits and the Na^+^/-H^+^ exchanger NHE1 and thus affects pH_i_ regulation [276,277]. However, as both Na_V_1.5 and NHE1 lead to an intracellular Na^+^ load, it is still unclear how the cells provide the necessary driving force for H^+^ efflux. 

The neonatal splice variant of Na_V_1.5 is more resistant to extracellular acidosis than the adult Na_V_1.5: under acidic pH_o_ conditions, voltage-dependent activation of the neonatal form is shifted less towards more depolarized voltages than that of the adult form [278]. This differential pH_o_-sensitivity may be relevant as the **neonatal Na_V_1.5** is normally not detectable in adult tissues. It becomes functionally expressed in metastatic breast and colon cancer, as well as in neuroblastoma cells, promoting tissue invasion and in vivo metastasis [184,279,280,281]. It remains to be established whether Na_V_ inhibitors with increasing activity at acidic pH, such as ranolazine, would have a beneficial effect in cancer therapy [282]. 

#### 4.4.2. Piezo1

Since its characterization in 2010 [283], the Piezo channel family has been extensively studied, as Piezo1 and Piezo2 are key mechanosensors in the human body. They are ubiquitously expressed and are able to sense multiple forms of mechanical stimuli and transduce them into appropriate responses depending on the cell type. For example, Piezo1 is vital for sensing vascular shear stress and regulating of red blood cell volume, whereas Piezo2 is crucial for proprioception (for greater detail see [284,285,286]).

The **Piezo1** channel enhances the invasive phenotype in multiple tumors, namely breast, colorectal and prostate cancers [193,287,288]. An electrophysiological study described that heterologously expressed Piezo1 channels are inhibited by extracellular protons [188]. These results have been confirmed and extended by our group. We showed channel inhibition also by an intracellular acidification in pancreatic stellate cells that are key stromal cells in pancreatic ductal adenocarcinoma [189]. According to our findings, the Piezo1 channel is likely to be in a nonconducting state in acidic, poorly perfused regions of solid tumors. This could protect the cells from cell death due to harmful Ca^2+^ overload. On the other hand, Piezo1 would be fully responsive to mechanical stimuli in well-perfused areas of the tumor, when it is relieved from the protonation-dependent channel inhibition.

#### 4.4.3. Acid-Sensing Ion Channels (ASICs)

Acid-sensing ion channels (ASICs) are members of the degenerin/epithelial sodium channel (ENaC) family and are widely expressed in the nervous system [289]. They are sensitive to extracellular acidosis and are involved in acid-induced nociception. In humans, four genes (*ASIC1-4*) give rise to at least six distinct subunits, [290]. ASICs are sensitive to inhibition by amiloride and are permeable to Na^+^ to a much greater extent than to K^+^ and Ca^2+^.

For a long period, ASIC channels remained enigmatic ”brain” Na^+^ channels with known ENaC homology. The discovery of H^+^-gating of these channels propelled subsequent research. Low extracellular pH (pH ~6.5 for ASIC1a and ASIC3; ~6.1 for ASIC1b and ~4.5 for ASIC2a) induces channel opening. The current is transient and followed by rapid channel desensitization [291]. ASIC3 and some heteromers do not fully desensitize, showing small, sustained currents [292]. ASIC3 activation by H^+^ results from a competition of Ca^2+^ and multiple protons at the activation site of the channel in a manner that low [Ca^2+^]_o_ (10 µM) allows channel gating at a pH close to physiological [43].

ASICs sense acidic pH in the tumor environment and are thereby involved in tumor progression [13]. The expression of functional **ASIC1** was shown in breast cancer cells, where it induces acidosis-induced ROS production, cancer invasion and growth. The underlying mechanism is possibly based on ASIC1-mediated Ca^2+^ influx [4]. ASIC1 and ASIC3 are overexpressed in pancreatic cancer and support acidosis-induced epithelial–mesenchymal transition (EMT). In this instance, the authors also suggest the Ca^2+^ permeability of **ASIC1/3** as a pivotal mechanism which leads to subsequent activation of RhoA signaling [194]. In colorectal cancer, **ASIC2** promotes metastasis and its expression correlates with poor prognosis [195]. In a nonsmall cell lung cancer line, A549, several ASIC members are expressed and mediate cell proliferation and migration [196]. Overall, based on these few studies, the role of ASIC channels in an acidic tumor environment appears to depend mainly on the Ca^2+^ permeability of the channel.

#### 4.4.4. P_2_X

Purinergic receptors are ATP-gated Na^+^-, K^+^- and Ca^2+^-permeable channels. There are seven receptor subunits described (P_2_X1-7), among which P_2_X1–4 are modulated by pH_o_ [293]. pH-induced inhibition of **P_2_X_7_** depends on acidification and duration of the low pH_o_ exposure [198,294]. Antagonists of this channel are already used in therapy of neuropathic pain, and there is evidence of an inhibitory impact on cancer growth and metastasis [295]. In melanoma, colon cancer and PDAC cell lines, P_2_X_7_ activity promotes tumor progression [197,200]. P_2_X_7_ receptor is gaining more attention, especially due to its expression in tumor-infiltrating immune cells [197,296], detailed in Section 5.1. Whether or not the pH-sensitive P_2_X1–4 channels are involved in cancer development still remains unclear.

## 5. pH Dependence of Ion Channels in Tumor Immunity

One aspect which the literature has consistently highlighted in orchestrating a powerful immune reaction is that immune cells require intact ion channels in the cell membrane. Nondisruption of those ion channels is pivotal for the immune system’s ability to efficiently carry out its function of destroying cancer cells. This applies to cells of both the innate and adaptive immunity.

### 5.1. Innate Immunity

Cells of innate immunity make up an important part of the tumor milieu. Tumor-associated neutrophils and macrophages, dendritic cells (DC) and myeloid-derived suppressor cells (MDSC) are potential targets of immune-based therapies [297,298,299,300]. Although the innate immune response is generally associated with leukocytes of myeloid lineage, descendants of the lymphoid lineage such as natural killer (NK), natural killer T-cells (NKT) and γδ T cells also fall into this category [301]. A high number of myeloid cells in the TME and in the blood circulation predominantly correlates with poor prognosis [302]. In contrast, effector cells of lymphoid lineage (such as NK cells) display beneficial antitumorigenic characteristics [303]. However, different tumors and varied immunomodulatory microenvironments often require tailored approaches.

Low extracellular pH of solid tumors is an important modulator of the immune response. Several Ca^2+^ permeable, pH-sensitive ion channels are expressed in neutrophils, macrophages and DCs, including STIM/Orai, TRP channels and purinergic P_2_X receptors. These channels, detailed below and in Table 1, are generally inhibited by protons. Inhibition of the Ca^2+^ fluxes in the acidic tumor environment may, therefore, result in the retention of compromised (protumorigenic) immune cells in the tumor microenvironment and ultimately in disease aggravation.

Infiltrating neutrophils are unequivocally *cellae non gratae* in varied tumors [302,304]. Neutrophil migration and recruitment are based on CXCR2 signaling and require concurrent activity of the **TRPC6** channel [139]. Inhibition of the TRPC6 channel by extracellular protons may hinder neutrophil chemotaxis. **TRPM2**, which is inhibited at low extra- and intracellular pH, and **TRPM7**, may be involved in neutrophil migration [305,306,307,308,309]. However, the underlying mechanisms and consequences of channel inhibition in neutrophils are not well described to date. Other neutrophil functions such as production of ROS and release of neutrophil extracellular traps also rely on undisturbed Ca^2+^ fluxes. Moreover, the activity of phagocyte NADPH oxidase NOX2 requires constant charge compensation which is provided by **H_V_1** [310]. As mentioned in Section 4.3, H_V_1 opens when there is a considerable H^+^ gradient across the plasma membrane. Therefore, an extracellular acidification may inhibit this channel and disrupt NOX2 activity and hence, impair the killing capacity of neutrophils.

In the case of macrophages, plasticity plays an important role in the TME. Macrophage polarization into the so-called M2 type, is one of the mechanisms for rendering the tumor milieu immunosuppressive [311]. The pH-sensitive TRP channels, namely **TRPM2, TRPM7 and TRPC1** are involved in modulating macrophage phenotype, and therefore, extracellular pH also indirectly controls this process [312,313,314].

The purinergic **P_2_X_7_** receptor is expressed in a variety of immune cells in the TME. For instance, P_2_X_7_ is involved in phagocyte migration and ROS production [315]. Targeting P_2_X_7_ in inflammation-related diseases is already implemented in clinical trials [316]. In dendritic cells, STIM/Orai -mediated Ca^2+^ fluxes are also pivotal for presenting antigens. Inhibition of STIM/Orai channels in dendritic cells in the acidic TME may contribute to the fact that low pH_o_ disrupts the activation of lymphocytes.

In addition, not-fully differentiated myeloid-derived suppressor cells (MDSCs) contribute to TME immune anergy. Two major populations resemble granulocytes and monocytes/macrophages and have strong immunosuppressive features [317]. In MDSCs, purinergic **P_2_X_7_** mediates CCL2 release, macrophage recruitment and contributes to MDSC expansion [201]. In the case of hepatitis, TRPV1 activity induced or potentiated by acidic pH_o_ stimulates MDSCs and results in the resolution of the inflammatory process [318]. Not much else is known about pH-sensitive ion channels expressed in MDSCs. It remains to be seen whether the immunosuppressive MDSC function can be overcome by channel modulation through alteration of the microenvironment.

### 5.2. Acquired Immunity

Ca^2+^ influx mediated by ion channels is a precondition for triggering lymphocyte metabolism, activation and a range of downstream signaling pathways [319].

Regarding metabolism, naïve T cells are in a “quiescent” state and dependent on oxidative phosphorylation as a source of energy [320]. Antigen-stimulated T cells shift their metabolism to aerobic glycolysis [321]. In this context, it is notable that T cell metabolism is also regulated by Ca^2+^ signaling through the Ca^2+^/calcineurin/NFAT pathway, which is involved in controlling the expression of many different elements involved in glycolysis, such as glucose transporters GLUT1 and GLUT3, glycolytic enzymes and transcription factors HIF1α, IRF4 and c-Myc [322]. There is a number of ways in which T cells can be inhibited in the acidic TME due to the fact that both cancer cells and activated T cells use aerobic glycolysis as their primary form of metabolism. For example, having high amounts of lactate production (from the cancer cells) in the TME prevents activated T cells from effluxing lactate thus blocking the glycolytic metabolism in the T cells [323]. Cancer cells and immune cells must, therefore, compete for the available glucose which is limited in the established tumor region. Whilst cancer cells are highly adaptable, T cells are unable to adapt and subsequently enter an anergic state [321,324].

T lymphocyte activation occurs when the T cell antigen receptors (TCRs) recognize and bind antigens, with this antigen–receptor engagement eliciting a response from the phospholipase Cγ (PLCγ)/inositol 1,4,5-triphosphate (IP_3_) pathway which provokes Ca^2+^ release from the endoplasmic reticulum (ER). The eventual depletion of those Ca^2+^ stores stimulates the opening of Ca^2+^-release activated Ca^2+^ channels (CRAC) in the plasma membrane [325,326]. The substantial increase in [Ca^2+^]_i_ activates NFAT transcription complexes which are used to assemble and mediate the transcription of the crucial genes involved in T cell activation, such as IL-2. Rapid Ca^2+^ influxes, however, cause depolarization of the membrane potential, which, if not regulated, would act to prevent a further build-up of Ca^2+^. This is prevented by the counterbalancing efflux of potassium ions out of the cell. This K^+^ efflux, mainly mediated by the (pH-sensitive) voltage-gated K_V_1.3 and Ca^2+^-activated K_Ca_3.1 channels, is essential in maintaining the hyperpolarized membrane potential state which is in turn pivotal in sustaining the Ca^2+^ influx [68,326].

As observed from the examples above, ion channels orchestrating the initiation, intensity and duration of the Ca^2+^ signal are critical for an effective immune response, also in the case of tumor immunity. With respect to the antigen–receptor engagement for T lymphocytes, the **CRAC (ORAI-STIM)** channels constitute the dominant pathway involved in mediating the store-operated Ca^2+^ entry (SOCE) regulated Ca^2+^ influx [188,327]. Opening of the CRAC channels occurs by the STIM proteins binding to the Orai subunits that form the Ca^2+^ selective conducting pore. Depending on the activation and the subtype of T lymphocytes, K_V_1.3 channels or K_Ca_3.1 channels are the main channels involved in regulating the membrane potential in order to sustain SOCE [328].

The general pH-sensitivity of **K_V_ and K_Ca_** channels is detailed in Section 4.1, where we already pointed out that an acid-induced inhibition of the K^+^ conductance might be compensated by an upregulation in the respective channels (in lymphocytes), with the result that the net K^+^ conductance is maintained [49]. Consequently, one reason for K^+^ remaining in the T lymphocytes is due to this pH blockage of the efflux of K^+^. Additionally, the increased concentration of K^+^ in the interstitial fluid leads to an increased intracellular concentration of K^+^ in T cells. The elevated extracellular K^+^ concentration, in turn, is due to the fact that in both human and mouse tumors, cell death (necrosis or apoptosis) leads to the release of high levels of K^+^ ions into the interstitial fluid, with this action impairing the effector function of T cells [92]. The higher concentration of K^+^ in the T cells inhibits the transcription of the genes involved in stimulating the Akt-mTOR pathway (a central regulator of cell growth and metabolism) which is heavily involved in regulating the effector T cell function. Overexpressing K_V_1.3 in tumor-specific T cells lowers intracellular K^+^ concentrations by increasing K^+^ efflux into the interstitial fluid and thereby improves the functioning of the effector T cells by increasing Akt-mTOR activation and the production of IFN-γ [92].

**CRAC** channel activity is reduced by both intra- and extracellular acidic pH [329,330]. For the pH sensitivity of Orai1, Glu190 and Glu106 residues are pivotal when Na^+^ or Ca^2+^ are the main current contributors, respectively [166,168]. Many studies have shown that blocking/deleting Orai1 attenuates the SOCE process in T cells [331]. Age is associated with a marked decrease in the expression of the Orai and STIM proteins in the CD8^+^ T cells which reduces Ca^2+^ current through the Orai channels [332]. This decrease in Ca^2+^ signaling is linked to the resistance of cytotoxic lymphocyte (CTL) cytotoxicity efficiency. This can be an explanation for the preponderance of tumors with age. It also suggests a reduced efficiency of the antitumorigenic CD8^+^ response to tumor cells, if Ca^2+^ influx is consistently blocked by the acidic TME. Many of CTL’s mechanisms of killing tumor cells are dependent on Ca^2+^ flux [333]. A study proposed a bell-shaped dependence of CTL cytotoxicity on Ca^2+^, where partial inhibition of Orai1 in CTLs decreases Ca^2+^ signaling but improves the cytotoxicity of the T cells [181]. Whether such partial inhibition can be achieved therapeutically by fine-tuning the environmental pH of the TME is a subject of further studies.

Other pH-sensitive Ca^2+^ permeable channels such as members of the TRP family (detailed in Section 4.2) also contribute to Ca^2+^ influx in lymphocytes [325]. **TRPM7** is important in T cell development and functioning [334]. TRPM7 channels can be impaired by the presence of protons. One study shows that TRPM7 channels, functionally cooperating with K_Ca_3.1, are localized at the rear part (uropod) of migrating T lymphocytes and regulate cell migration. Inhibiting K_Ca_3.1 and downregulating TRPM7 hinders T cell motility. It is suggested that this process is executed through the regulation of Ca^2+^ influx, with K_Ca_3.1 maintaining the membrane potential and with the TRPM7 channel facilitating the influx of Ca^2+^ [335]. As acidic pH_i_ and pH_o_ inhibit TRPM7 function, the channel is not likely to be functional in lymphocytes residing in acidic areas of tumors, resulting in impaired cell migration. **TRPC1 and TRPC3**, which are involved in Ca^2+^ entry in T lymphocytes, have yet to be investigated in this aspect [336,337,338,339]. **TRPV1 and TRPV4** are also involved in Ca^2+^ signaling in T cells [340]. Being activated by both extra- and intracellular protons, they would enable cation influx and possibly Ca^2+^ signaling in an acidic environment. Overall, the TRP family members contribute to many of the crucial pathways needed in T cell activation and function. Yet, it needs to be further studied how pH may affect their activity and function.

A particular focus in this section was on the role of ion channels in T lymphocytes. However, more information is needed on the different subpopulations of T cells affected by these channels. The expression of K^+^ channels—such as K_V_1.3 and K_Ca_3.1-as well as CRAC and other ion channels—varies in different T lymphocyte types and cell subsets also depending on their state of activation. Similarly, evidence of the involvement of cation channels in the effective functioning of B lymphocytes is still scarce. Many of the ion channels discussed above show some evidence of overlapping in B cell function, namely the role of CRAC channels in Ca^2+^ influx [326]. The proton channel H_V_1 promotes B cell signaling, and B cells deficient in H_V_1 have impaired ROS generation, which in turn affects their metabolism, modulating mitochondrial respiration and glycolysis [341].

To summarize, dysregulated pH impairs T-cell activation and T-cell function. The acidic TME may block the CRAC and K^+^ channels in tumor-infiltrating immune cells. Diminished Ca^2+^ influx into the T cell in turn blocks the crucial dephosphorylation of NFAT, thereby preventing the production of several cytokines, such as IL-2 and IFN-γ, which are needed for effective T cell functioning [323]. Furthermore, intratumoral acidosis through lactate could prevent the expression of the pivotal proteins needed for the metabolism of activated T cells. In addition, there may be an overlap of metabolic dysfunctions, which are acting together with the acidic pH_o_ to dampen the T cell function. Hypoxia-driven metabolic dysfunctions cause adenosine accumulation in the extracellular tumor microenvironment and together with acidic pH_o_ could negatively affect T-cell function through the immunosuppressive effects of adenosine A_2A_ receptors [342,343]. If this hypothesis is correct, then it could suggest an overlap with K_Ca_3.1, which is adenosine sensitive and required for T cell motility. Along these lines, blockage of K_Ca_3.1 could contribute to causing T cell anergy in the acidic pH_o_ [119,120,342,343].

## 6. Conclusions

Studying pH and pH-dependent processes in cancer is a timely task. Its relevance becomes particularly evident when considering that some of the cancers with the worst prognosis originate from tissues with an extreme acid-based homeostasis. Prominent examples include gastric and pancreatic cancer whose host organs generate enormous acid-based fluxes after each meal that in turn cause marked changes in the interstitial pH. To understand how these alterations in pH can ultimately push cells into forming a malignant tumor and favor its progression is a prerequisite for developing new pH-oriented diagnostic as well as therapeutic options. To achieve this, a lot of groundwork is still required such as intense pH-imaging at the cellular, tissue and organism levels.

Such information is indispensable for decoding the role of pH-sensitive ion channels in the crosstalk between the cancer cells, cells of the tumor immunity and the components of the TME. Remarkably, the family of ion channels shows a large heterogeneity on how they respond to different pH. For example, most K^+^ channels are inhibited in an acidic environment typical for hypoxic, poorly vascularized areas inside a tumor, thereby affecting every cell type in this environment. Whether this acidic pH-dependent inhibition of channels results in T cell anergy and ineffective innate tumor immunity still remains an open question. On the other hand, cation flux is probably maintained by an upregulation of many acid-inhibited channels and by a number of channels that are activated by an acidic environment (for example, TRPV1). In this way, Ca^2+^ signals needed for cell migration to escape the unfavorable acidic environment could still be generated.

In summary, our review was aimed at promoting studies that deepen our understanding of the complex pH landscape of the tumor microenvironment and to understand the role of ion channels in cancer in a more holistic view. This is indeed one of the major conclusions of our review. There is still a large gap of knowledge in understanding the mutual interplay between ion channels in cancer and the pH landscape of solid tumors. The latter applies not only to the channels themselves but how their biophysical properties confer mutual dependencies. There is still a lot to learn about ion channels regulating the function of the different cell types that make up a tumor. Intriguingly, we frequently find the same channel types in different cells of the tumor microenvironment. For example, K_Ca_3.1 channels are expressed not only in tumor cells but also in stromal and endothelial cells as well as in tumor-infiltrating immune cells such as lymphocytes, macrophages and neutrophils. Complex tumor models are needed in order to disclose the therapeutic benefits or side effects of such a “multiple-hit” ion channel-targeting therapy. While this is certainly a challenging endeavor, we should keep in mind that ~15% of all clinically prescribed drugs are already targeting ion transport proteins, many of which can be repurposed for use in cancer therapy. Larger inter-/national consortia should be created to advance this exciting research field further.

## Figures and Tables

**Figure 1 cancers-12-02484-f001:**
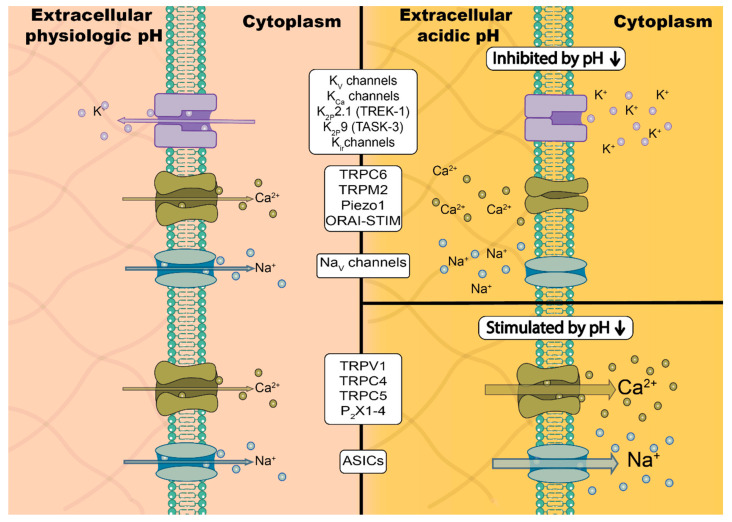
Response of ion channels to an acidic environment. The left side indicates physiological interstitial pH condition (pH_o_ = 7.4), whereas the right side depicts an acidic environment prevalent in tumors (pH_o_ = 6.5). The size of the arrows represents ion flux rate.

**Table 1 cancers-12-02484-t001:** pH-dependence of ion channels involved in cancer.

Gene Name	Channel Name	Cancer Type	Functions in Cancers	pH_o_ Acidification	pH_i_ Acidification
*KCNA3*	K_V_1.3	colorectal (cell line)	↑ migration, adhesiveness [88]	slight inhibition [89,90]	
		breast (cell line)	↑ migration, adhesiveness, proliferation [88,91]		
		melanoma (T lymphocyte)	↑ antitumor immunity in tumor-bearing mice [92]		
		head and neck (T lymphocyte)	defects are associated with diminished cytotoxicity [93]		
*KCNA5*	K_V_1.5	lung (cell line, tumor)	apoptosis, H_2_O_2_ production, ↓ tumor growth [27]	strong inhibition, pK_a_ = 7.2 [94]	inhibition [27,95]
		cervical (cell line)	apoptosis, H_2_O_2_ production [95]		
		gastric (cell line)	↑ proliferation [96]		
*KCNB1*	K_V_2.1	uterine (cell line)	↑ cell cycle progression [97]	slight inhibition [94]	
*KCNQ1*	K_V_.7.1	colorectal (cell line)	↑ proliferation [98]	slight inhibition, pK_a_ = 5.5 [99]	inhibition, BUT stimulation in presence of KCNE1 [100]
*KCNH1*	K_V_10.1 (hEAG1)	glioblastoma (cell line, tumor)	↑ tumor growth and metastasis [101]	inhibition, pK_a_ ≈ 6.5 [102]	
		colorectal (cell line)	↑ proliferation [103]		
		gastric (cell line)	↑ proliferation [104]		
		breast (cell line, tumor)	↑ tumor progression and hypoxia signaling [105]		
		osteosarcoma (cell line)	↑ proliferation, cell cycle progression [106]		
		osteosarcoma (endothelium)	knockdown inhibitis tumor angiogenesis and VEGF-signaling [107]		
*KCNH2*	K_V_11.1 (hERG1)	pancreatic (cell line)	↑ EGFR-signaling, proliferation, migration [85]	slight inhibition, pK_a_ = 5.1 [45,108]	slight stimulation [109]
		gastric (cell line, tumor)	↑ PI3K/Akt pathway, promotes angiogenesis, proliferation, migration and tumor growth [110]		
*KCNN4*	K_Ca_3.1	glioblastoma (cell line, tumor)	↑ migration, swelling, invasion, radiotherapy resistance, tumor growth (reviewed in [111]	possibly insensitive between pH 6.5 and pH 8 [112]	inhibition [113]
		breast (cell line, tumor)	↑ cell cycle progression, proliferation and tumor growth [114,115]		
		pancreatic (cell line)	↑ migration, invasion, proliferation [116]		inhibition [113]
		pancreatic (stromal cell)	↑ migration, chemotaxis [117]		
		melanoma (T lymphocyte)	↑ antitumor immunity in tumor-bearing mice [92,118]		
		head and neck cancer (T lymphocyte)	↑ tumor infiltration [119,120]		
*KCNJ1*	K_ir_1.1	renal (cell line)	↓ cancer cell growth, invasion and promotes cancer cell apoptosis [121]	strong inhibition pK_a_ = 6.9 [122]	
*KCNJ10*	K_ir_4.1	glioblastoma (cell line)	↓ cell cycle progression [123]	inhibition pK_a_ = 6 [124]	
*KCNK2*	K_2P_2.1 (TREK1)	prostate (cell line)	↑ proliferation, contributes to setting the membrane potential [125]	strong inhibition pKa = 7.35 [126]	slight stimulation [127]
		pancreatic (cell line)	↑ proliferation, migration, contributes to setting the membrane potential [128]		
*KCNK3, KCNK9*	K_2P_3.1(TASK1), K2P9.1(TASK3)	glioblastoma (cell line)	↑ apoptosis and hypoxia resistance, contribute to setting the membrane potential [129]	inhibition [126]	
		lung (cell line)	↑ proliferation, contribute to setting the membrane potential [130]		
*KCNK5*	K_2P_5.1(TASK2)	breast (cell line)	↑ proliferation, contribute to setting the membrane potential [131]	inhibition [132]	inhibition [132,133]
*TRPC4, TRPC5*	TRPC4, TRPC5	renal (cell line)	activation leads to cancer cell death [134]	stimulation [135]	
*TRPC6*	TRPC6	glioblastoma (U373MG cell line)	↑ cancer cell proliferation, invasion, angiogenesis [136]	slight inhibition [135]	
		liver (cell line)	↑ proliferation [137]		
		pancreatic (tumor-associated immune cell)	↑ CXCL1/CXCR2 induced neutrophil infiltration [138,139]		
		pancreatic (stromal cell)	↑ migration, hypoxia response [140]		
*TRPM2*	TRPM2	breast (cell lines and murine model)	mediates H_2_O_2_ cytotoxicity [141,142]	inhibition pKa = 6.5 [143]	complete inhibition by pH = 6 [143]
		prostate (cell lines)	↑ proliferation [144]		
*TRPM7*	TRPM7	nasopharyngeal (cell lines)	↑ migration [145]		inhibition pKa = 6.32 [146]
		lung (A549 cell line)	↑ migration [147]		
		pancreatic (cell line)	↑ migration [148]		
		breast	↑ proliferation and migration (overexpressed in cancer tissue) [149]		
*TRPV1*	TRPV1	breast (cell line)	apoptosis, tumor suppressor (mRNA overexpression) [150]	stimulation [151]	
		gastric and colorectal (cell line)	apoptosis, tumor suppressor (decreased expression in cancer tissues) [152]		
		astrocytoma (murine model)	apoptosis, tumor suppressor [153]		
		endometrial (cell line)	apoptosis, tumor suppressor [154]		
*TRPV4*	TRPV4	lung (murine model)	↓ endothelial tumor cell migration, normalization of tumor vascularization [155]	stimulation [156]	
		breast (tumor-derived EC, cancer cell line)	↑ promotes tumor-derived endothelial cell migration and metastasis [157]		
		hepatocellular carcinoma (cell line and murine model)	TRPV4 inhibition leads to cell apoptosis [158]		
		gastric (cell line)	↑ cancer proliferation, migration and invasion [159]		
*HVCN1*	H_V_1	breast (cell line, tumor)	promotes cell migration, tumor growth, pH_i_ regulation [160,161]	pH_o_/pH_i_ relation and membrane-potential dependent, detailed in Section 4.3	
		colorectal (cell line)	promotes cell migration, pH_i_ regulation [162,163]		
		glioblastoma (cell line)	promotes cell migration, survival, pH_i_ regulation [164]		
		B-cell lymphoma (patient samples, cell line)	short variant of H_V_1 promotes cell migration and proliferation [165]		
	CRAC channel (ORAI-STIM)			inhibition pKa = 8.3 [166]	iinhibition [166]
*ORAI1/STIM1*	CRAC channel	renal (primary cells)	↑ migration and proliferation [167]	inhibition pKa = 8.3 [168]	inhibition pK_a_ = 7.5 [166]
		breast (primary cells, cell line)	↑ migration, invasion [169]		
		esophageal (cell line)	↑ migration and proliferation [170]		
		glioblastoma (primary cells, cell line)	↑ migration, invasion, proliferation and ↓ apoptosis [171,172]		
		lung (cell line)	↑ proliferation [173]		
		melanoma (cell line)	↑ proliferation, migration and invasion; metastasis formation [174]		
		pancreatic (cell line)	apoptosis resistance [175]		
		gastric (cell line)	↑ proliferation, migration, invasion [176]		
		acut myeloid leukemia (cell line)	↑ proliferation and migration [177]		
		liver (cell line)	↑ proliferation [178]		
		colorectal (primary cells, cell line)	↑ proliferation, invasion and metastasis [179]		
		colorectal (T lymphocyte)	↑ tumor cell killing by CD8^+^ T cells, regulates antitumor immunity [180]		
		melanoma (T lymphocyte)	↑ tumor cell killing by CD8^+^ T cells, regulates antitumor immunity [180]		
		leukemia (T lymphocyte and NK cell)	↓ tumor cell killing due to verly high Ca^2+^ signals [181]		
*ORAI2/STIM1*	CRAC channel	acut myeloid leukemia (cell line)	↑ proliferation and migration [177]	inhibition pK_a_ = 8.3 [168]	
*ORAI3/STIM1*	CRAC channel	breast (cell line)	↑ proliferation [182]	inhibition pK_a_ = 8.5 [168]	
		prostate (primary cells, cell line)	↑ tumor growth, apoptosis resistance [183]		
*SCN5A*	Na_V_1.5	breast (cell line, biopsies)	↑ migration, invasion and cancer progression [33,184,185]	inhbition pK_a_ = 6.1 [186]	
*PIEZO1*	Piezo1	gastric (cell line)	↑ proliferation, migration, chemotherapy resistance [187]	inhibition [188,189]	inhibition [189]
		synovial (cell line)	↑ viability and migration [190]		
		gliobastoma (cell line)	↑ cell proliferation [191]		
		breast (cell line)	↑ invasion, cell migration [192]		
		prostate (cell line)	↑ proliferation, migration [193]		
		pancreatic (stromal cell)	↑ migration and invasiveness [189]		
*ASIC1*	ASIC1	breast (cell line and murine model)	↑ invasion, metastasis and ROS production [4]	stimulation (reviewed in [44])	
*ASIC1, ASIC3*	ASIC1/3	pancreatic (cell lines)	↑ epithelial–mesenchymal transition [194,171]		
*ASIC2*	ASIC2	colorectal (cell line and murine model)	↑ cancer proliferation, invasion and metastasis [195]		
	ASIC1a	lung (cell line)	↑ proliferation and migration [196]		
*P2X7*	P2X7	melanoma (murine model)	↑ tumor growth, reduced apoptosis, increased angiogenesis [197]	inhibition pK_a_ = 6.7 [198]	
		cervical cancer (cell line)	↑ cancer cell apoptosis [199]		
		pancreatic (cell line)	↑ tumor progression [200]		
		neuroblastoma (tumor-associated immune cell)	promotion of MDSCs; CCL2 release, macrophage recruitment [201]		

↑ indicates increase, whereas ↓ indicates decrease of a given function.

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
