# Peer review of "pH-Channeling in Cancer: How pH-Dependence of Cation Channels Shapes Cancer Pathophysiology"

_cancers, 2020, doi:10.3390/cancers12092484_

Round 1
Reviewer 1 Report
The authors have choosen an intriguing topic, i.e. the role of pH-sensitive ion channels in cancer progression, in particular in the crosstalk between cancer cells themselves or with other cellular components of the tumor microenvironment. Describing the function and the role of pH-sensitive ion channels in cancer and disease progression, the authors often reported a heterogeneous behaviour, underling the need of further studies on this field. The review is well written and comprehensible. Some minor points:
Lane 331: please revise the sentence
Lane 534: please correct “inder”
Lane 623: please, better introduce Piezo1 channel (e.g., include more details about its physiological function)
Lane 694: please, include appropriate references and revise punctuation.
Lane 722-723: please, add appropriate references
Lane 849-856: besides the role on tumor development, this would be the appropriate section in this manuscript to briefly describe the role of the acidic TME on tumor progression (EMT, metastasis development, anoikis resistance, chemoresistance …; please refer to detailed literature for instance on melanoma and breast cancer progression)
Author Response
Replies to Reviewers’ comments
pH-channeling of cancer development (Review)
Reviewer 1.
The authors have choosen an intriguing topic, i.e. the role of pH-sensitive ion channels in cancer progression, in particular in the crosstalk between cancer cells themselves or with other cellular components of the tumor microenvironment. Describing the function and the role of pH-sensitive ion channels in cancer and disease progression, the authors often reported a heterogeneous behaviour, underling the need of further studies on this field. The review is well written and comprehensible. Some minor points:
- Lane 623: please, better introduce Piezo1 channel (e.g., include more details about its physiological function)
We would like to thank the Reviewer for the helpful comments. Indeed, the introduction regarding Piezo channels fell short compared to other channel families. To correct this issue, we supplemented the introductory part of section 4.4.2. with additional information and references for better orientation (lane 689-694):
“Since its characterization in 2010 [304] the Piezo channel family has been extensively studied, as Piezo1 and Piezo2 are key mechanosensors in the human body. They are ubiquitously expressed and are able to sense multiple forms of mechanical stimuli and transduce them into appropriate responses depending on the cell type. For example, Piezo1 is vital for sensing vascular shear stress and regulating of red blood cell volume, whereas Piezo2 is crucial for proprioception (for greater detail see [305–307]).”
- Lane 849-856: besides the role on tumor development, this would be the appropriate section in this manuscript to briefly describe the role of the acidic TME on tumor progression (EMT, metastasis development, anoikis resistance, chemoresistance …; please refer to detailed literature for instance on melanoma and breast cancer progression)
We agree that a more thorough description on the role of the acidic tumor microenvironment on cancer progression supports the readability of the manuscript. As the Introduction briefly mentioned this topic, we found it even more beneficial to highlight it there. Among the multiple aspects that are changes by the acidic TME, we emphasized the ones that later reappear in the review (for instance: invasiveness, metastasis development, immunosuppression) and hence not explicitly mentioned e. g. EMT and chemoresistance. However, as suggested, we referred to comprehensive reviews on multiple tumors that include other aspects too. We supplemented lanes 118-123 with the following:
„Acidosis in the TME is linked to tumor progression via a large variety of factors including among others: extracellular matrix remodeling; the promotion of invasion and metastasis by affecting cell adhesion; and it instigates many immunosuppressive processes which result in a significant dampening of the immune response. These phenomena have been widely discussed in comprehensive reviews in multiple types of cancer, e. g. breast cancer [4,5], melanoma [6–8] and pancreatic cancer [9].”
Lane 331: please revise the sentence
Lane 534: please correct “inder”
Lane 694: please, include appropriate references and revise punctuation.
Lane 722-723: please, add appropriate references
Corrected. We aimed to correct further formal and punctuation issues as well and restructured the manuscript to make it more understandable and easier to read. We marked those changes with the “track changes” tool in Word to make them easier to follow.
Reviewer 2 Report
The authors present a timely, well-written and interesting review about the importance of proton-sensitive ion channel function in cancer development. Overall, a well-referenced and useful review that will be a valuable resource.
Minor points are a few spelling errors:
Line 103: patter --> pattern
Line 366: most the existing --> most of the existing studies
Line 399: through the protonation a conserved glutamate --> through the protonation of a conserved glutamate
Line 414: unclear
Line 425: the introduction of an loss-of-function --> the introduction of a loss-of-function
Line 534: taking inder consideration
Line 555: Multiple ion channels that are capable of conducting H+, as mentioned in previous sections, the most specialized channel being the voltage-gated proton channel HV1 --> Multiple ion channels are capable of conducting H+, as mentioned in previous sections, the most specialized channel being the voltage-gated proton channel HV1.
Line 617: as the neonatal NaV1.5, normally not detectable --> as the neonatal NaV1.5, normally is not detectable
Author Response
Replies to Reviewers’ comments
pH-channeling of cancer development (Review)
Reviewer 2.
The authors present a timely, well-written and interesting review about the importance of proton-sensitive ion channel function in cancer development. Overall, a well-referenced and useful review that will be a valuable resource.
Minor points are a few spelling errors:
Line 103: patter --> pattern
Line 366: most the existing --> most of the existing studies
Line 399: through the protonation a conserved glutamate --> through the protonation of a conserved glutamate
Line 414: unclear
Line 425: the introduction of an loss-of-function --> the introduction of a loss-of-function
Line 534: taking inder consideration
Line 555: Multiple ion channels that are capable of conducting H+, as mentioned in previous sections, the most specialized channel being the voltage-gated proton channel HV1 --> Multiple ion channels are capable of conducting H+, as mentioned in previous sections, the most specialized channel being the voltage-gated proton channel HV1.
Line 617: as the neonatal NaV1.5, normally not detectable --> as the neonatal NaV1.5, normally is not detectable
We are thankful to the Reviewer for the positive feedback. We have corrected all spelling errors pointed out and further formal issues (e.g. in references) in the manuscript. All changes were marked with the “track changes” tool in Word to make them easier to follow.
Reviewer 3 Report
This is a comprehensive review summarizing the role of cation channels in the tumor cell function. The authors acknowledged relevant studies in the field and provided detailed review of major findings in the field. The level of details is greatly appreciated, yet the manuscript is very dense in content of the broad cancer field that make it difficult to convey in a particular direction and justify its novelty as compared to other reviews in the same topic. This review will benefit of refining the content to focus in a sub-topic of current interest in the cancer field such as the field of tumor immunity or a particular therapeutic approach.
Major comments:
- This review is very dense and provides a lot of details that make it difficult to convey in a particular direction. Perhaps, this review will benefit of re-focus in a particular niche such as “pH channeling in carcinogenesis and tumor immunity” (as several sections have this focus already) rather than in the broad cancer field.
- It is not clear how this review is different from other general reviews in the same topic including:
- Cancer as a channelopathy: ion channels and pumps in tumor development and progression doi: 3389/fncel.2015.00086
- Ion Channels in Cancer: Are Cancer Hallmarks Oncochannelopathies? https://doi.org/10.1152/physrev.00044.2016
Minor comments:
- In Section 4, the text should come before the caption and figure.
Author Response
Replies to Reviewers’ comments
pH-channeling of cancer development (Review)
Reviewer 3.
This is a comprehensive review summarizing the role of cation channels in the tumor cell function. The authors acknowledged relevant studies in the field and provided detailed review of major findings in the field. The level of details is greatly appreciated, yet the manuscript is very dense in content of the broad cancer field that make it difficult to convey in a particular direction and justify its novelty as compared to other reviews in the same topic. This review will benefit of refining the content to focus in a sub-topic of current interest in the cancer field such as the field of tumor immunity or a particular therapeutic approach.
Major comments:
- This review is very dense and provides a lot of details that make it difficult to convey in a particular direction. Perhaps, this review will benefit of re-focus in a particular niche such as “pH channeling in carcinogenesis and tumor immunity” (as several sections have this focus already) rather than in the broad cancer field.
We are grateful to the Reviewer for the insightful comment and pointing out the need for restructuring. The review handles the niche of “pH-dependence of ion channels” in regard to cancer, but the amount of details in the manuscript may have indeed blurred the direction we wanted to emphasize. To correct this issue, we have performed the following:
- We further pinpointed the aim of our present work in the Introduction section, lanes 141-149 :
„Ion channels have a key role in cancer progression affecting every hallmark of cancer, as pointed out in numerous comprehensive reviews, e. g. [12–15]. However, our critical review of the literature revealed that most of the studies investigating the role of ion channels in cancer do not consider the pH landscape of tumors. Therefore, the present review specifically emphasizes the complex interplay pH and individual ion channels in cancer pathophysiology. We propose multiple mechanisms how ion channel function and interaction with the cell adhesion machinery in cancer cells and in tumor immunity can be affected by pH. We discuss possible pathophysiological consequences of pH-dependent ion channel activity and highlight the need of further studies in this direction.”
- We reworked the manuscript extensively rooting out peripheral details distracting from the main message. We omitted parts that were: A) not directly relevant for cancer, for instance lanes 330-331 in section 4.1; B) general biophysical characteristics of ion channels, e. g. lane 611-614 in section 4.3; C) already mentioned in other parts of the review or could be referred to appropriate references in the broad cancer field (for example lanes 373-376 or 851-852). The changes are too numerous to list them all here, but we tracked each modification with the “track changes” tool in Word. This way the review became almost 2000 words shorter that hopefully also aids readability.
II. It is not clear how this review is different from other general reviews in the same topic including:
Cancer as a channelopathy: ion channels and pumps in tumor development and progression doi: 3389/fncel.2015.00086
Ion Channels in Cancer: Are Cancer Hallmarks Oncochannelopathies? https://doi.org/10.1152/physrev.00044.2016
We are thankful for the suggested references which were previously mistakenly not included in our manuscript. We agree that there are multiple reviews published every year handling ion channels/transporters/pumps in cancers. However, our primary aim is to highlight that most ion channels are very strongly modulated by the pH of the tumor microenvironment, but we do not yet know how this affects cancer behavior. We included a sentence in the Conclusion on this matter, lanes 953-955:
„In summary our review was aimed at promoting studies that deepen our understanding of the complex pH landscape of the tumor microenvironment and to understand the role of ion channels in cancer in a more holistic view. This is indeed one of the major conclusions of our review. There is still a large gap of knowledge in understanding the mutual interplay between ion channels in cancer and the pH landscape of solid tumors.”
To underline the “pH-dependence” focus that also distinguishes our manuscript from other reviews, we described in the Introduction how acidosis of the tumor microenvironment is linked to cancer development (lanes 68-73). To further stress the focus on ion channels above transporters and pumps, we have removed parts focusing on transporters/pumps, for instance lanes 133-135. Moreover, we emphasized the unique features of our manuscript compared to the previous reviews.
For instance, lanes 314-318:
„…protonation by an acidic environment often results in a loss-of-function of cation channels involved in cancer cells and thereby to an inhibition of ion fluxes, similarly to what Prevarskaya et al. proposed [13]. This general phenomenon, however, shows a huge variability depending on how strongly pH-sensitive a channel is, ranging from almost insensitive to markedly pH-dependent.”
Minor comment:
- In Section 4, the text should come before the caption and figure.
Corrected.
Round 2
Reviewer 3 Report
The authors have restructure the content of the review and carefully addressed all the comments.
Author Response
We are thankful for the Reviewer for handling our manuscript and for the positive reply.